# Design of Military Service Framework for Enabling Migration to Military SaaS Cloud Environment

**Sungrim Cho [1], Sunwoong Hwang [2], Woochang Shin [3], Neunghoe Kim [4] and Hoh Peter In [4,\***

[1] Korea Institute for Defense Analyses and Department of Computer Science and Engineering, Korea University, Seoul 02841, Korea; srcho@korea.ac.kr
[2] Department of Military Digital Convergence, Ajou University, Gyeonggi-do 16499, Korea; swhwang@ajou.ac.kr
[3] Department of Computer Science, Seokyeong University, Seoul 02713, Korea; wcshin@skuniv.ac.kr
[4] Department of Computer Science and Engineering, Korea University, Seoul 02841, Korea; nunghoi@korea.ac.kr
\* Correspondence: hoh_in@korea.ac.kr

**Abstract:** The military of the Republic of South Korea applies infrastructure as a service (IaaS) cloud-computing technology and applications distributed nationwide are transferred to two defense-integrated data centers (DIDCs). However, to improve the quality of the military information service provided by the DIDCs, it is crucial to expand the cloud-computing service from IaaS to the software as a service (SaaS) level. Owing to the in-house planning, development and operation of military applications by the Ministry of National Defense and each military force, these organizations operate several applications that are similar and redundant. Accordingly, SaaS can reduce the costs of IT services and management overhead by eliminating similar and redundant applications and integrating them into common services. In this study, a military service framework (MSF) was designed based on the criteria of business functions and service targets. The MSF was applied to applications currently in operation to create a current military SaaS portfolio, whereas the future military SaaS portfolio was restructured by identifying common services within and between organizations and throughout the military. Consequently, 369 future military SaaS portfolios were created, representing a 36% decrease realized by integrating similar and redundant systems out of 580 applications.

**Keywords:** SaaS; cloud computing; portfolio; data center; military application





## 1. Introduction

Government agencies and public institutions are constantly seeking ways to reduce operating costs and IT management overhead, as well as improve service quality for the public. Accordingly, cloud computing is garnering significant attention as one of the measures to realize this goal [1,2]. Cloud computing comprises shared-resource and elastic computing services offered on demand, which are pooled and accessible via external data centers and the Internet, respectively [3].

The military of the Republic of South Korea (ROK) is faced with several operational challenges, such as the nationwide distribution of numerous applications, wastage of budget and personnel resources and exposure to various disasters and security threats owing to insufficient infrastructure. To address these challenges, the Ministry of National Defense (MND) integrated 240 computing centers nationwide and two defense integrated data centers (DIDCs). Owing to the operation of DIDCs, application failures decreased by 34% compared to the previous year and the ability to counter cyber threats increased by 23%, thus improving the IT service environment. Currently, the MND is implementing a "cloud-first" policy that would increase the cloud application rate from 15% to more than 60% by 2021.

The cloud-computing technology applied to DIDCs is infrastructure as a service (IaaS), which focuses on the distribution of hardware infrastructure, such as servers, storage and

networks. However, to improve the quality of the military information service provided by DIDCs, the cloud-computing service needs to be expanded from the IaaS level to the software as a service (SaaS) level. Currently, because the MND, Army, Navy and Air Force plan, develop and operate their own applications separately, these organizations operate numerous applications that are similar and redundant. SaaS can reduce IT service costs and management overhead by discarding similar and redundant applications, integrating them into common services and managing them in a single military SaaS portfolio.

In this study, a military service framework (MSF) was designed to systematically migrate military applications to the SaaS cloud environment and manage the SaaS portfolio. The proposed framework was designed under several preconditions considering the ROK military's cloud policy. One of such preconditions was applying a private cloud deployment model and another was migrating military applications to a SaaS cloud environment. This study does not include the contents of SaaS security.

Therefore, by implementing the proposed MSF, military information service developers can avoid duplicated developments and provide novel services using existing services. In addition, the proposed MSF is expected to reduce the maintenance costs of military applications, thereby significantly lowering the barriers to the acquisition of new technologies for military applications.

The remainder of this paper is organized as follows. In Section 2, the trends in cloud-computing technology, as well as strategies for adopting cloud computing in major countries are reviewed and in Section 3, the MSF design is described. Section 4 provides an example of the MSF application for managing the SaaS portfolio. Finally, Section 5 presents the summary and implications of this study, as well as future research directions.

## 2. Research Background and Trend Analysis

According to the market trends of SaaS released by the Synergy Research Group, the SaaS market is growing at approximately 30% annually with software vendors realizing more than $23 billion in sales in the enterprise SaaS market in the first quarter of 2019 [4]. Accordingly, the governments in several countries are adopting cloud-computing systems and conducting projects to convert government applications into SaaS.

### 2.1. Cloud-Computing Trend

According to the National Institute of Standards and Technology (NIST), "cloud computing is a model for enabling convenient, on-demand network access to a shared pool of configurable computing resources (e.g., networks, servers, storage, applications and services) that can be rapidly provisioned and released with minimal management effort or service provider interaction [5]." Cloud computing has three delivery models: IaaS, SaaS and platform as a service (PaaS), as well as four deployment models—public, private, community and hybrid clouds.

The concept of cloud computing enables users to access information and data at any time and from anywhere without restrictions or hardware equipment requirements [6]. Cloud services can contribute to the flexibility of the computing infrastructure of an organization, increase its capacity to deliver service solutions and reduce IT infrastructure costs, regardless of an increase in demand [7].

However, there are also security threats associated with cloud computing [8]. The Cloud Security Alliance (CSA) continuously reports on possible security threats in the cloud environment [9]. The thirteen major threats reported include: (1) data breaches, (2) poor identity, authentication and access management, (3) insecure interfaces and APIs, (4) system vulnerabilities, (5) account hijacking, (6) malicious insiders, (7) advanced persistent threats, (8) data loss, (9) insufficient due diligence, (10) abuse and malicious use of cloud services, (11) denial of service, (12) shared technology vulnerability and (13) meltdown and specter [10].

Despite these cloud security threats, governments and public institutions still seek to reduce costs, minimize IT management overhead and provide innovation and other

benefits. With cloud computing, these institutions can reduce IT capital expenditures via pay-per-use subscriptions and improve their services to the public, as well as overcome challenging financial crises [3,11,12].

However, public organizations still lag behind private organizations in adopting cloud-computing services. Presently, private companies are introducing SaaS services in typical common business functions, such as enterprise resource planning, customer relationship management and supply chain management [13,14]. The SaaS environment enables enterprises to access software via the Internet without on-premise installations, while avoiding high costs for initial installations [12]. The SaaS concept has been heralded as a novel method for software service provisioning because it enables the faster implementation of software changes and allows organizations to eliminate their IT installation, control and maintenance [15,16]. Consequently, several organizations are beginning to implement cloud-based solutions as replacements for their legacy systems [17].

However, despite the increasing adoption of SaaS in the public sector, many countries face several challenges in the initial stage of adopting SaaS [13]. In particular, because SaaS providers supply a wide range of software products, their integration with legacy systems and data security pose major challenges [18–20].

The ROK military is building a private cloud-computing environment in which DIDCs are physically separated from the outside world by prioritizing military security. This study focused on applying the SaaS model to DIDCs to provide IT services for the military.

### 2.2. Cloud-Computing Adoption in Major Countries

In 2019, the US government announced a federal cloud-computing strategy [21]. The core of this strategy is the cloud-first policy, which implements various cloud-computing service models, such as private, public and hybrid clouds and has reduced the IT costs of the federal government from $80 billion to $20 billion. However, by adopting cloud computing, the expenses of the US government have increased significantly each year; hence, a novel strategy is needed to achieve additional cost reduction, security and swift service delivery based on the implementation of the cloud-first strategy.

In the report presented to the President on "Federal IT Modernization," which was released publicly in 2017, the Office of Management and Budget pledged to update the federal cloud-computing strategy, cloud first [22]. To fulfill this promise, a novel strategy, cloud smart, was developed to accelerate the agency's adoption of cloud-based solutions. This strategy considers security, procurement and human resources as the basis for the successful adoption of cloud computing. Additionally, according to the plan, all federal agencies are required to rationalize their application portfolio to drive the adoption of the federal cloud. The rationalization process is expected to reduce the application portfolio by assessing the necessity and usage of applications and discarding obsolete, redundant or excessively resource-intensive applications [22].

In 2011, the British government announced the government's cloud strategy as part of its ICT strategy. Based on this policy, the government established the G-cloud strategy to introduce cloud services in the public sector and launched the G-cloud Store in 2012. The G-cloud Store is a procurement system in which cloud services are provided in the form of a catalog. Via this initiative, the challenge posed by the redundant development of IT services in each department has been addressed.

In 2013, the government announced a cloud-first policy to necessitate cloud computing when developing IT infrastructure in the public sector, which in 2017 evolved into the public cloud-first policy that prioritizes the civil cloud. The digital market place, which evolved from the cloud store, started with 1700 services in 2012 and currently offers 24,543 services. More than 78% of the public sector that adopted one or more cloud-computing services were estimated to have saved approximately £725 million in the fiscal 2016/17 [23,24].

In 2011, the government of Canada incorporated data centers, networks and e-mail into Shared Services Canada via cloud computing technology and in 2012, the government published a report on the state of IT obsolescence across the Canadian government [24]. In

November 2017, the Canadian government issued reports on the "Directions for the Safe Use of Commercial Cloud Services: Security Policy Implementation Notice" and "Directions for Electronic Data Residency." On February 7, 2018, the Canadian government declared at the GC Cloud First Day event that it would begin the procurement process for "protected cloud service." The right cloud strategy evolved into the cloud-first strategy, which is considered first for IT delivery, while public cloud has become the preferred model.

China is pursuing a cloud activation policy by announcing its Three-Year Cloud Development Action Plan (2017–2019) to increase the cloud market of China to $61.2 billion and strengthen corporate competitiveness by 2021 [24]. Japan promoted the Kasemigaseki Project in 2013 to support the introduction of cloud computing by central ministries and local governments and it established a plan for reducing operating costs by 30% via transferring all government applications to the cloud by 2021 [24].

The ROK has established a foundation for cloud computing, including the enforcement of the Cloud Computing Act (March 2015) [25], establishment of the first basic plan for government cloud computing (November 2015) [26,27], implementation of cloud security certification (May 2016) and guidelines for cloud computing (July 2016), as well as the establishment of the second basic plan (December 2018) [28]. According to the first plan, the government computing center integrated 1970 servers with 255 high-performance servers to build a pan-government cloud infrastructure and platform, research and develop core cloud technologies and establish a cloud service base for active introduction in the public sector. In 2019, the size of the domestic cloud computing market increased by 25.2% over the previous year to 1.3 trillion [29]. The reason for the lack of cloud activation in the public sector was attributed to concerns over information leakage and regulations that hinder cloud adoption [28].

*2.3. Cloud-Computing Adoption of the ROK Military*

The number of applications operated by the ROK MND increased from 347 before 2003 to 2186 in 2008. This increase in applications increases operating personnel and management costs, as well as cyber threats, such as hacking.

To address these challenges, the MND established a DIDC to consolidate all computing centers that are dispersed and operated by the computing centers of the Army, Navy and Air Force [30]. The DIDC enhanced the efficiency and utilization of military information resources and improved stability, as well as survivability. It also ensured business continuity by providing information services without interrupting the preparation for natural and social disasters.

In the first phase, the DIDC integrated 240 computing centers into 77 from 2003 to 2007. In the second phase, 77 computing centers were integrated into 2 by 2012 and IaaS cloud-computing technology was adopted to establish an operation infrastructure in 2014 [31]. In addition, similar or duplicate systems among existing applications were integrated or discarded via the military application transfer and integration project. Consequently, the military discarded or integrated 3215 applications into 1415. The military integrated reservation and administration functions into an intranet homepage, thereby discarding 1800 applications, transferring 1110 applications to the DIDC and leaving the remaining 281 applications.

## 3. Design of MSF

The MSF is a model used for presenting the direction in which military information services should be developed according to changes in future military environments. The MSF is necessary for visualizing information services to provide mission-oriented services and preventing redundant development by integrating the similar functions of all military forces.

In this section, the scope of the military information service is defined and the criteria for designing the MSF are established. Furthermore, the procedure for developing the SaaS portfolio by applying the MSF is defined. This procedure can be used to restructure the military's SaaS portfolio.

### 3.1. Scope of Military SaaS Cloud Service

The scope of the military SaaS cloud service in this research is limited to the business management applications in DIDC, as illustrated in Figure 1. In addition, it excludes the battlefield management applications that are based on different operating environments, such as computing centers and communication networks. The military SaaS cloud will be implemented by applying the MSF to the applications on the IaaS cloud adopted in DIDC.

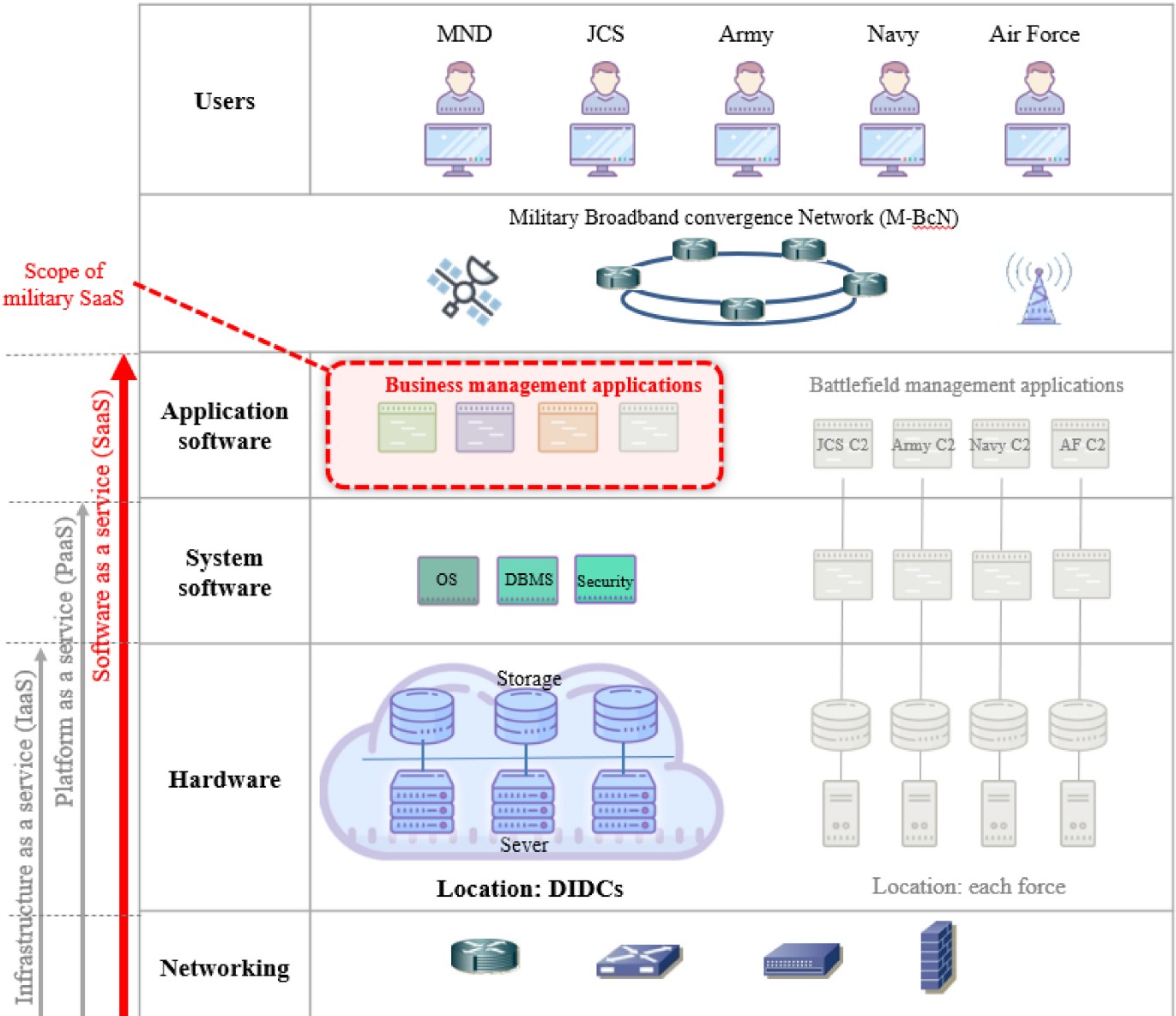

**Figure 1.** Military cloud service.

### 3.2. Design of MSF Structure

To design the MSF, we first analyzed the criteria for migrating applications were first analyzed. Military applications are classified according to the directive of the defense information service. This classification system is divided into four functions: planning and finance, personnel and mobilization, logistics and facilities and finally electronic administration based on military application. Subsequently, it is subdivided into 18 subfunctions. However, while this classification system can identify the work function of an application, it is limited to identifying the supervisor of the management or service target.

In addition, the MND has developed enterprise level applications for all four functional fields. Meanwhile the Army, Navy and Air Force have developed unique own applications for each functional field. This has resulted in redundant application development associated with an increase in development and maintenance costs. In addition, despite troop reductions, system management personnel must be maintained. Finally, interoperability issues arise between applications. Therefore, the military requires a framework that supports the development of individual applications for the unique work of each military force while simultaneously developing jointly enterprise level applications for similar work of each military force.

As investigated in Chapter 2 Related Studies [18,19], there are many difficulties due to integration with legacy systems and data security. This is especially true when introducing the SaaS cloud to defense applications. For security reasons, the ROK military is contemplating an approach to build a private SaaS cloud and migrate legacy systems.

The private SaaS cloud environment provides an environment for efficient development, operation and maintenance based on common services. Because it is not a public SaaS cloud environment, it has no relation with the cost of service but instead relates to the migration methodology of the existing military application. This final issue is important.

The ROK MND conducted a nationwide project to transfer legacy applications from 77 local computing centers to the DIDC [30]. This project classified applications into five categories: civil service, common, same, similar and unique systems. This classification system provides criteria for classifying applications by analyzing redundancy and similarity.

This classification system focuses on the similarity between the applications that are already in operation. Accordingly, although it has the advantage of reducing redundant applications, it is limited in its application as a standard for the systematic planning, operation and management of applications. The classification system of the applications can be used as a standard for: (1) managing applications continuously, (2) maintaining consistency with the existing classification system and (3) assessing the similarity between the applications, as well as integrating them to ensure their cost-effective management.

In this study, we analyzed the existing classification systems and then designed the classification system of the MSF suitable for transferring applications to the cloud environment. Military information services were divided into application and infrastructure services. Application and infrastructure services support business functions and application services, respectively.

Two criteria were considered for subdividing the application service. The first criterion is the classification by business function, which is the existing standard adopted by the MND to classify applications. In addition, the MND plans, develops and operates applications according to this classification system. As a standard, this criterion can be consistently and continuously applied when performing information service planning, development and operation tasks. As an axis of the MSF, it is divided into four major functions: planning and finance, personnel and mobilization, logistics and facilities and electronic administration.

The second criterion is classification by the service target organization. This criterion is determined according to the level of the organization receiving the service. In other words, there are unique services for specific organizations, as well as similar services that are operated for each organization but created with similar functions. Finally, enterprise services exist for two or more organizations or for all troops. The organization unit referred here comprises the MND, Army, Navy and Air Force, followed by the inclusion of numerous suborganizations under each military unit. Classification by service target can identify the subject that develops and operates the applications and the organizations that receive the service. In addition, it clarifies the subject of management accordingly. Figure 2 illustrates the structure of the MSF.

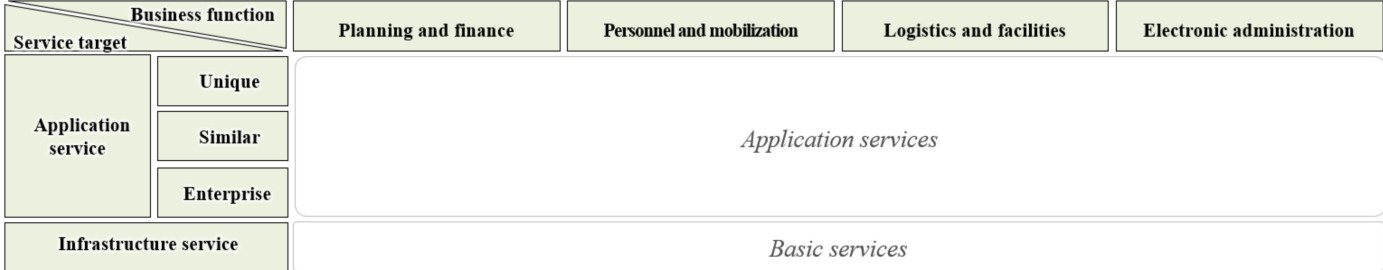

**Figure 2.** Military service framework (MSF) structure.

### 3.3. Identification and Classification of the Current Military SaaS Portfolio

By applying the classification system of the MSF, a SaaS portfolio was obtained from the currently implemented military applications. The derived SaaS portfolio has become a service element that comprises the MSF.

The procedure for identifying and implementing the military service cloud (SaaS) consists of three steps, as presented in Table 1. In the first step, the latest status of applications subject to military SaaS are investigated, followed by the reclassification of applications according to business functions and service targets, which comprise the criteria of the MSF. Accordingly, the current military SaaS portfolio is defined.

**Table 1.** Procedure for the creation of military software as a service (SaaS) portfolio.

| Steps | Activities | Output |
|---|---|---|
| 1. Target investigation | ① investigate the latest status of information systems<br>② apply the MSF<br>③ define the current defense SaaS portfolio | Current military SaaS portfolio |
| 2. Identification of common services | ① identify common services within an organization (unique service)<br>② identify common services between organizations (similar, enterprise service)<br>③ identify infrastructure services | Common service candidates |
| 3. Definition of the military SaaS portfolio | ① set the target year<br>② define the future defense SaaS portfolio<br>③ analyze effort | Future defense SaaS portfolio |

In Step 2, the common services that should be provided as cloud services in the current military SaaS portfolio are identified. Common services are identified among the unique services operated by the MND and each military force. Common services are then identified among similar services operated by several organizations simultaneously. Finally, basic services, which support application services for the entire organization, are identified.

Step 3 defines the future military SaaS portfolio. First, the target year of the portfolio is set, a future military SaaS portfolio is then created with the common service candidates identified in Step 2 and finally, the effects of these actions are analyzed.

The 580 military SaaS portfolio created by applying the MSF can be summarized as illustrated in Figure 3, which presents only representative types of applications.

The MND and each military force have developed and operated unique services that support their own tasks, including similar services that support similar tasks. Although the MND operates enterprise services to support the different aspects of the entire military, such as finance, personnel and logistics, no basic service has been recently developed and operated to support application services, such as unique, similar and enterprise services.

| Category | | Planning and finance | Personnel and mobilization | Logistics and facilities | Electronic administration |
|---|---|---|---|---|---|
| **Application service** | **Unique** | **Planning** | **Personnel** | **Logistics** | **Promotion** |
| | | Analysis & evaluation results management | Recruitment unit deployment result | Automated warehouse management | Low Carbon Green Growth |
| | | New weapon system evaluation | Submarin crew recruitment | Integrated Logistics Support | AirForce Traditional Hall of Fame |
| | | Test evaluation data manatement | Sound exploration career management | Clothing management | Naval Scholarship Foundation |
| | | Inspection activity management | Prisoner management | e-Clothing management | Ground Forces Festival |
| | | Legal information integration | Overseas dispatch support | Army Art management | Security Education Support |
| | | Cyber crime report | Wartime manpower operation | Airforce Art management | **Security** |
| | | Investigation Information Management | Employment support management | Tanker history management | Air Force Base Access Control |
| | | Performance management | **Education** | Logistics business management | naval access control |
| | | Organization data management | Aviation mechanic evaluation | Waste resource return management | Security reward management |
| | | **Finance** | Education and training materials | Foreign purchase management | Soldier equipment management |
| | | Meal history settlement | Education information | Logistics material management | Daily security management |
| | | Fund management | Cyber education | **Facility** | Storage media management |
| | | Overtime management | Technical manual management | Runway pavement maintenance | Electronic data leakage prevention |
| | | Travel expenses management | Leadership Diagnosis | Training ground management | Data exchange |
| | | Military employee salary management | **Mobilization** | Communication facility management | **Administration** |
| | | **Policy** | Preliminary military strength management | Tactical Heliport Management | Flight data management |
| | | Military diplomacy | wartime emergency recruitment | Water source management | Certificate Issuance |
| | | International liaison work | **Health welfare** | Environmental facility management | On-nara |
| | | **Informatization** | Welfare facility reservation | Environmental information | Performance management |
| | | Computer operation supplies management | Aviation medical management | Runway pavement management | Records management |
| | | SW development management | fitness center management | Renewable energy management | Knowledge management |
| | | Development analysis support | Mental health care | **Procurement** | History Hall Management |
| | | Program Management | Navy welfare potal | unit procurement management | e-mail |
| | **enterprise** | **Planning** | **Personnel** | **Logistics** | |
| | | Organizational capacity management | Military personnel management | Integrated logistics information | |
| | | Statistics DB | **Health welfare** | Equipment Maintenance | |
| | | **Finance** | Medical management | **Transport** | |
| | | Financial information management | Infectious Disease Information | Transportation Information | |
| | | Payroll management | **Mobilization** | **Facility** | |
| | | **Informatization** | Mobilization information | Facility management | |
| | | Information resource management | **Health welfare** | **Procurement** | |
| | | Interoperability Portal | Military pension | e-procurement | |
| **infrastructure service** | | *None* | | | |

**Figure 3.** Example of military SaaS portfolio by applying the MSF.

*3.4. Restructuring of Military SaaS Portfolio*

In this section, common services in the current military SaaS portfolio are identified and the SaaS portfolio for establishing the MSF is reconstructed. Therefore, to identify common cloud services in the current military SaaS portfolio, common services within the organization (unique services), between organizations (similar services or enterprise services) and throughout the organization (basic services) were identified.

The criteria for the identification of common services compare the duplication of or similarity between services and they consider the effects and economics that appear when consolidating common services. Services identified as common services are developed as new services or integrated with existing services.

3.4.1. Identification of Common Services within the Organization (Unique Services)

In this subsection, we describe the process of identifying common services within an organization. For example, the Air Force has developed and currently operates several applications related to inspection tasks, such as audit result management, inspection activity status management, safety inspection result management, integrated safety homepage, safety inspection performance management and safety recommendation management. These services correspond to services that support their own in-house businesses (i.e., unique services of the Air Force) and can be integrated into the Air Force supervision safety management service, considering service similarity and management efficiency, as illustrated in Figure 4. Service similarity was determined by considering the similarities between the names of applications, similarities between business functions and number of units using the service. By consolidating six services into one, management efficiency can reduce operation and maintenance personnel.

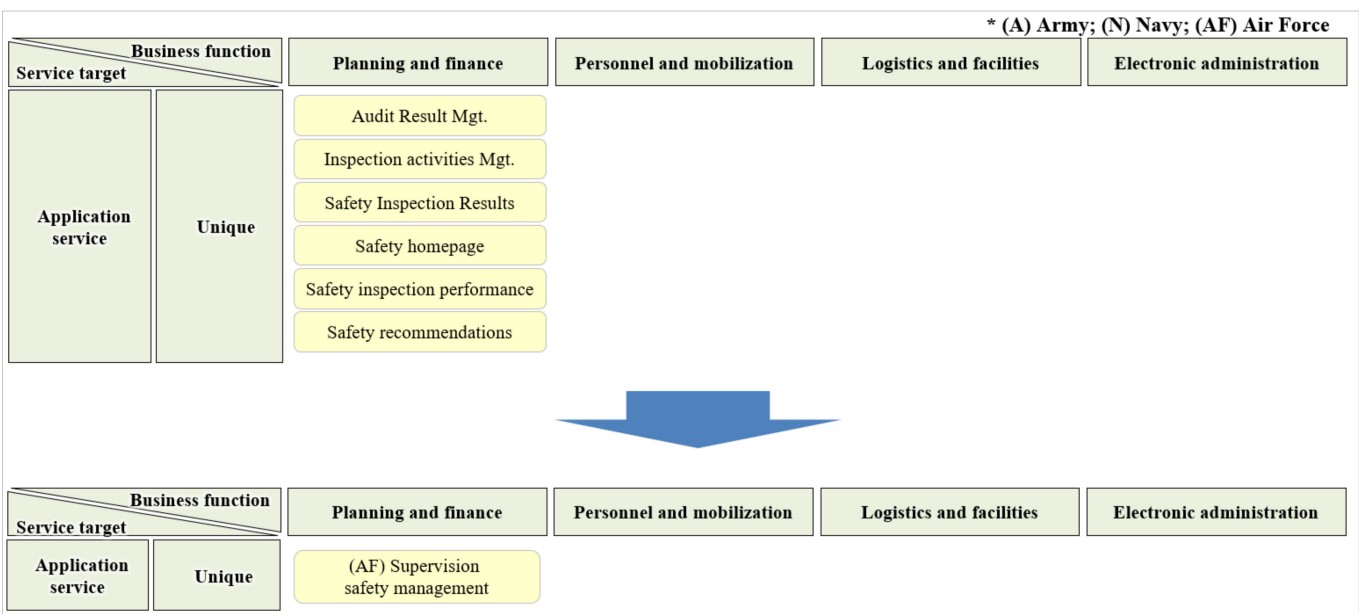

**Figure 4.** Example of integration of unique services in the Air Force.

### 3.4.2. Identification of Common Services between Organizations (Similar Services and Enterprise Services)

The Army, Navy and Air Force have developed and operated several applications with similar business functions. Each application is classified as a similar service provided it is required to be operated redundantly by each military organization. Furthermore, if it is necessary to expand the service to the entire military, it is developed as an enterprise service.

First, the identification of common services between organizations involves suggesting candidates for common services in the relevant organization, including the Army, Navy or Air Force. Second, an IT expert group verifies the target of common services among the candidates. The results can then be summarized and explained, as presented in Figure 5.

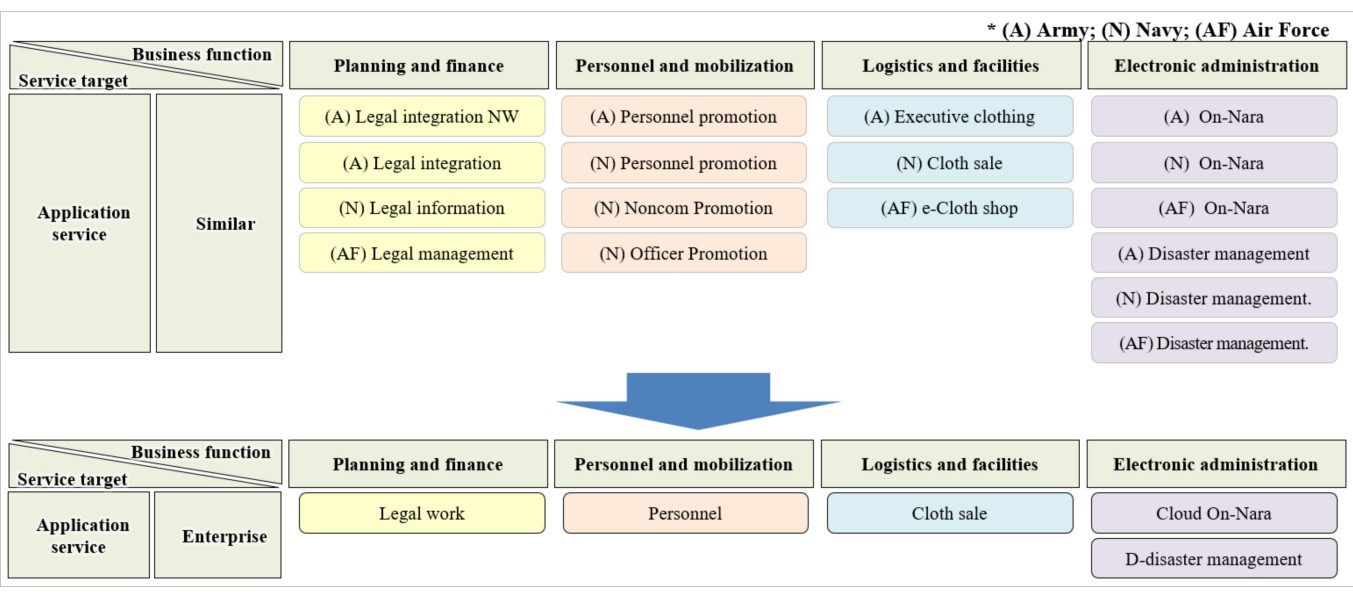

**Figure 5.** Cases of integration of similar services between organizations.

### 3.4.3. Identification of Common Cervices throughout the Organization (Basic Services)

Basic services are differentiated from application services to support the military's business. These services support the work of the user, such as knowledge management, collaboration tools and e-mail or can be used in conjunction with application services, such as integrated authentication and user management. The Army, Navy and Air Force have independently developed and operated applications corresponding to infrastructure services, including knowledge management, e-mail and user authentication. Figure 6 illustrates the integration of these services into infrastructure services used to support the entire military.

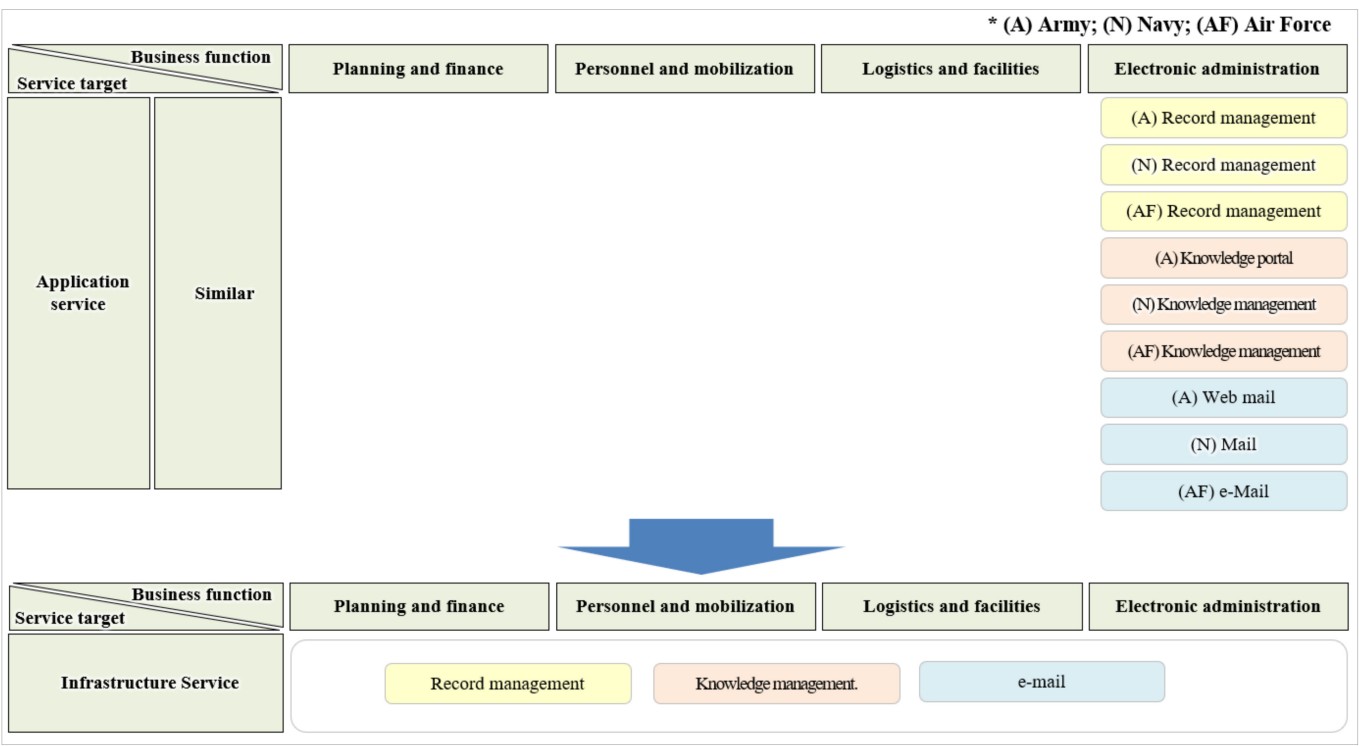

**Figure 6.** Example of basic service identification.

### 3.4.4. Definition of the Future Military SaaS Portfolio

The target year for the future military SaaS must be set. The target year determines how long it will take for the SaaS portfolio, which must be provided to all military users by the MND, to be prepared. This period is set considering the medium-term plan period of the MND (from F + 2 to F + 6).

### 3.5. Future MSF Restructured Based on SaaS Portfolio

The future MSF was designed by reconstructing the current SaaS portfolio per the procedure presented in Table 1. Figure 7 illustrates the resulting future MSF. Applying the MSF enables the systematic management of the future military SaaS portfolio. MSF migrates existing military applications to services; consequently, this creates a private SaaS cloud in the Korean military's data center. IT system, operation personnel and maintenance costs will be reduced. These savings will promote the further development of new services.

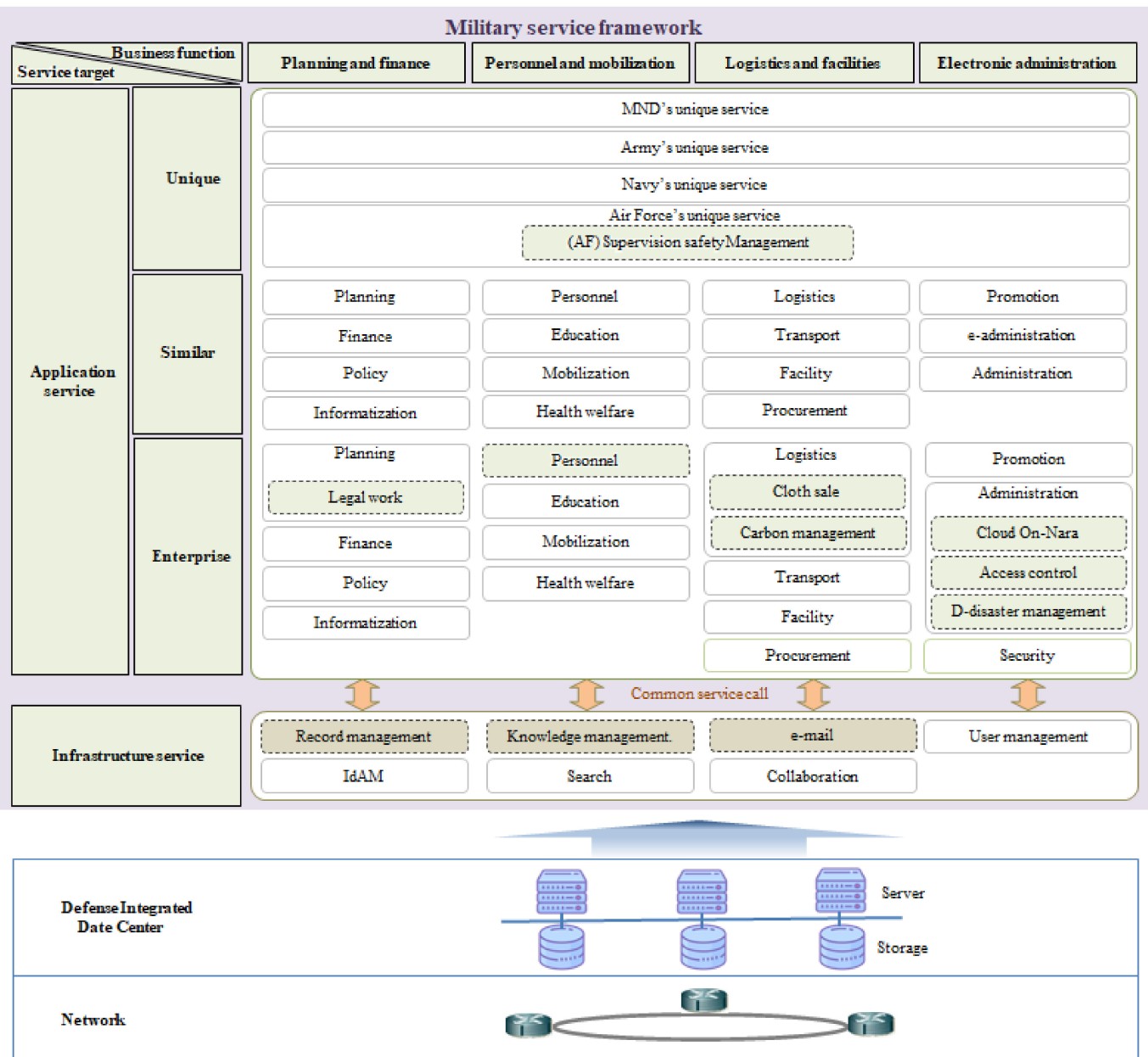

**Figure 7.** Restructured future MSF.

## 4. Example of the MSF Application

In the first step, the status of the applications operated by the MND and each military force was investigated. As of the year-end of 2019, the MND operated 834 military applications. Excluding battlefield management applications, 580 of them are eligible for the military service cloud. Therefore, the classification system of the MSF illustrated in Figure 7 was applied to 580 applications. They are classified according to business function and service target.

Table 2 presents the military SaaS portfolio classified by applying the MSF. Among the application services, 283 unique services, 268 similar services and 29 enterprise services were classified but no basic service was classified.

**Table 2.** Summary of current military SaaS portfolio.

| Category | | Planning and Finance | Personnel and Mobilization | Logistics and Facilities | Electronic Administration | Sub Total |
|---|---|---|---|---|---|---|
| **Application Service** | Unique Service | 47 | 70 | 41 | 125 | 283 |
| | Similar Service | 18 | 106 | 41 | 103 | 268 |
| | Enterprise Service | 8 | 5 | 8 | 8 | 29 |
| | Sub total | 73 | 181 | 90 | 236 | 580 |
| **Infrastructure Service** | | | | 0 | | |
| **Total** | | | | 580 | | |

In the second step, (1) unique services that provide services solely to the relevant organization were identified; (2) common services between organizations were identified and similar and enterprise services were classified; and (3) infrastructure services that support the entire military were identified. When identifying common services, similarity, effectiveness, obsolescence and development plans were considered. In each military force, the person in charge of information tasks identified and recommended common service candidates. Then, the MND policy officer and IT expert group reviewed them secondarily and confirmed these candidates as common services.

The case of identification of common services within the organization was previously described as the Air Force supervision safety management work presented in Figure 4. In the Air Force supervision work case, six services were integrated into a single service. Therefore, redundant services were integrated among unique services, thus reducing their number from 26 to 6. Consequently, the total number of unique services decreased from 283 to 263, as indicated in Figure 8.

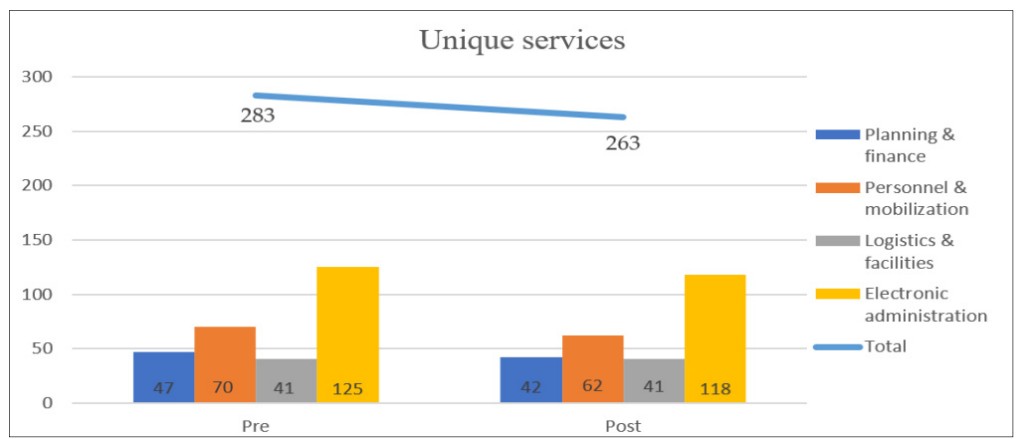

**Figure 8.** Unique services after applying MSF.

Similar services between organizations are redundantly operated services by each group. However, when the services are not required to be shared with other groups, they are integrated into organizational units as similar services. As presented in Figure 9, the Army, Navy and Air Force operate several applications related to the access control of military units included in the administration service. In particular, five Air Force military units of the Air Force operate separate access control services. Because access control services are not required to be shared with other forces, nine access control services are integrated into three services for the Army, Navy and Air Force, respectively. Although

it is a similar service between organizations, it is integrated into organizational units, as shown in Figure 9, because information sharing between organizations is unnecessary.

| | Category | | Electronic administration |
|---|---|---|---|
| | | | * (A) Army; (N) Navy; (AF) Air Force |
| | | | Administration |
| <Pre> | Application service | Similar | (A) Guardhouse access control |
| | | | (A) Scientific access control |
| | | | (N) Navy access control |
| | | | (N) English access system for Koreans |
| | | | (AF) Control unit base access management |
| | | | (AF) Access control |
| | | | (AF) Subordinate unit base access management |
| | | | (AF) Flight waiting line access control |
| | | | (AF) Military base access application |

| | Category | | Electronic administration |
|---|---|---|---|
| | | | Administration |
| <Post> | Application service | Similar | (A) Access control |
| | | | (N) Access control |
| | | | (AF) Access control |

**Figure 9.** Identification of similar services in the current SaaS portfolio.

As a result of integrating similar services between organizations in the manner described above, the number of similar services decreased from 268 to 38 as shown in Figure 10.

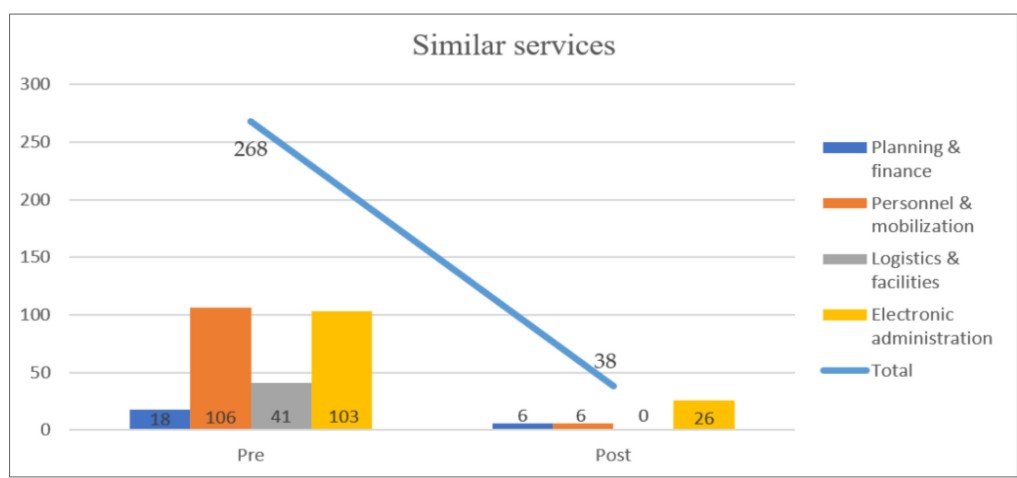

**Figure 10.** Similar services after applying MSF.

Among similar services between organizations, there is a common service that all military forces should utilize. These similar services are integrated into enterprise services. Figure 11 is an example of integrating similar services related to the planning and finance function to create a new enterprise service.

| | Category | | Planning and finance |
|---|---|---|---|
| | | | * (A) Army; (N) Navy; (AF) Air Force |
| <Pre> | Application service | Similar | Planning |
| | | | (A) Legal integration network |
| | | | (A) Legal integration system |
| | | | (N)  Legal integration system |
| | | | (AF) Legal management system |
| | | | Finance |
| | | | (A)  Investigation information management system |
| | | | (A)Scientific investigation system |
| | | | (N) Military police information system |
| | | | Informatization |
| | | | (A) Cybercrime report |
| | | | (N) Report and consultation |
| | | | (AF) Report and consultation |

| | Category | | Planning & finance |
|---|---|---|---|
| <Post> | Application service | Enterpise | Planning |
| | | | Legal work management |
| | | | Finance |
| | | | Investigation work management |
| | | | Informatization |
| | | | Cybercrime report and consultation |

**Figure 11.** Integrating similar services into new enterprise service.

Figure 12 shows an example in which similar services related to mobilization function are integrated into the existing mobilization system.

| | Category | | Personnel and mobilization |
|---|---|---|---|
| | | | * (A) Army; (N) Navy; (AF) Air Force |
| <Pre> | Application service | Similar | Mobilization |
| | | | (A) Reserve power management selection |
| | | | (A) Reserve power management deliberation |

| | Category | | Personnel & mobilization |
|---|---|---|---|
| <Post> | Application service | Enterprise | Mobilization |
| | | | (legacy) Mobilization system |

**Figure 12.** Integrating similar services into legacy enterprise service.

The number of enterprise services increased from 29 to 60, as shown in Figure 13. This increase was a result of integrating similar services between organizations into enterprise services, similar to the method presented previously. The number of enterprise services has increased independently from the unique service and similar services described above because similar services were integrated into enterprise services.

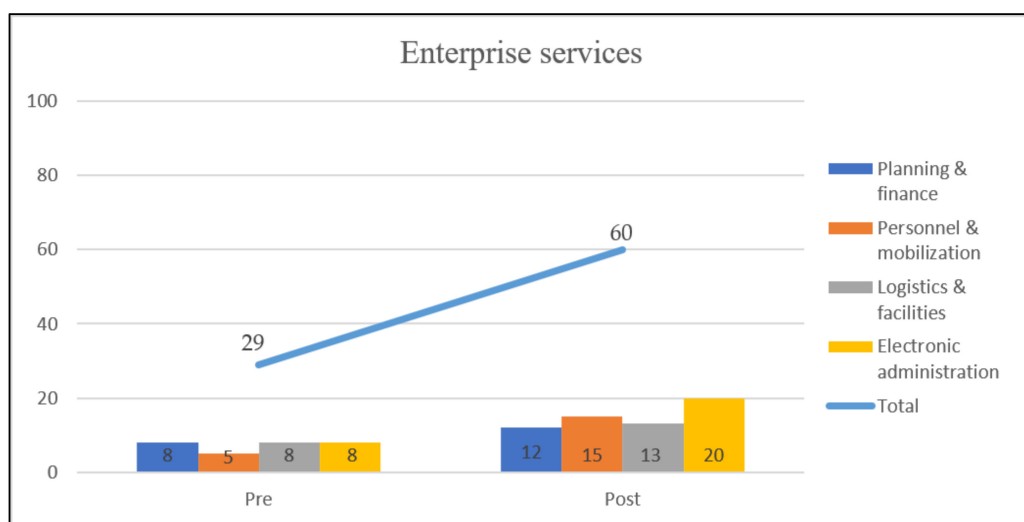

**Figure 13.** Similar services after applying MSF.

In the past, the MND defined services corresponding to the entire military infrastructure system under the name: "Common Operating Environment" (COE). However, although the concept of COE was designed, it is currently not in service. While designing the MSF, the service that was once defined as the COE was redefined in this study as the basic service in the infrastructure services. Infrastructure services include basic services, such as knowledge management, identity and access management (IdAM), e-mail, integrated search, total user management, mobile services, collaboration and smart office. The infrastructure service identified new services, increasing from 0 prior to the application of MSF to 8 after the application of MSF. This is shown in Figure 14.

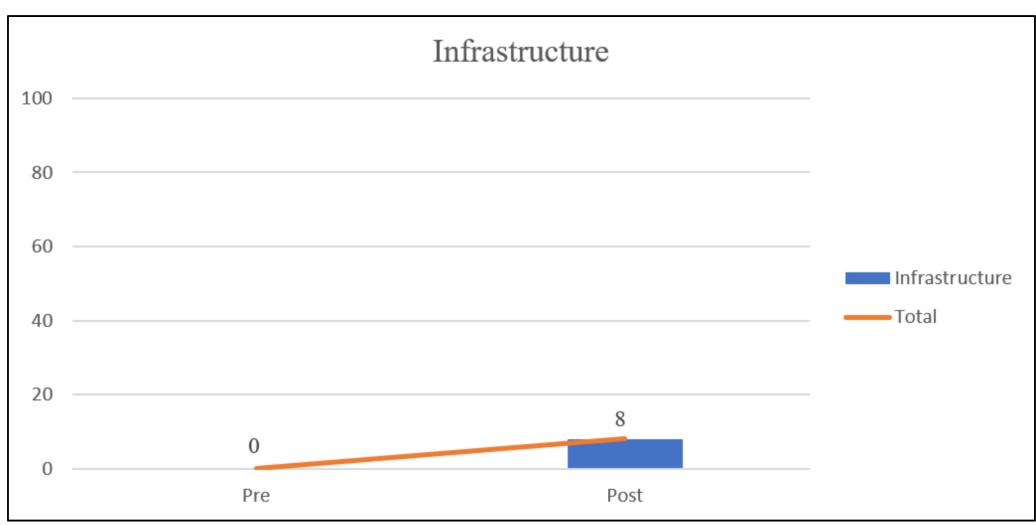

**Figure 14.** Infrastructure services applying MSF.

In step 3, the target year of the future military SaaS portfolio was set and the expected effects were analyzed. The future military SaaS target year was set according to the midterm plan of the MND (from F + 2 to F + 6), with 2020 as its starting point F; therefore, the target year was set to 2026, which corresponds to F + 6.

The applications currently operated by the MND were reorganized into a future military SaaS portfolio by applying the MSF. Table 3 summarizes the future military SaaS portfolio with the application of MSF. Consequently, 580 applications in the current portfolio of Table 2 were reclassified into 369 services in the future portfolio. Accordingly, 211 services, equivalent to 36%, were eliminated, thereby reducing service operation costs and operating personnel. These savings can be used to develop new services. In addition,

the new services may be more productive by including common services such as user authentication of infrastructure.

**Table 3.** Summary of the future military SaaS portfolio.

| Category | | Planning and Finance | Personnel and Mobilization | Logistics and Facilities | Electronic Administration | Sub Total |
|---|---|---|---|---|---|---|
| **Application Service** | Unique Service | 42 | 62 | 41 | 118 | 263 ($\bigtriangledown$20) |
| | Similar Service | 6 | 6 | 0 | 26 | 38 ($\bigtriangledown$230) |
| | Enterprise Service | 12 | 15 | 13 | 20 | 60 ($\bigtriangleup$31) |
| **Infrastructure Service** | | | | 8 ($\bigtriangleup$8) | | |
| **Total** | | | | 369 ($\bigtriangledown$211, 36.38%) | | |

## 5. Conclusions

In this study, we defined the MSF and application procedure and proposed a method for managing the future military SaaS portfolio by applying it to the current applications in operation. The proposed MSF categorized applications based on business functions and service targets. The business functions were divided into planning and finance, personnel and mobilization, logistics and facilities and electronic administration. In addition, it was made compatible with existing businesses based on previous application classification criteria. Furthermore, the service target was classified into unique, similar and enterprise services.

The ROK military currently applies IaaS cloud-computing technology and transfers applications distributed nationwide to two DIDCs. However, to improve the quality of the military information service provided by the DIDCs, it is necessary to expand the cloud-computing service from the IaaS to the SaaS level. Because the MND and each military force plan, develop and operate their own applications separately, several similar and redundant applications are operated in different organizations. SaaS can reduce IT service costs and management overhead by discarding similar and redundant applications and integrating them into a common service.

The future military SaaS portfolio was reconstructed based on the MSF proposed in this study. Based on business functions and service targets, 580 applications were reclassified and the future military SaaS portfolio was reorganized into 369 services. Consequently, 211 services, equivalent to 36%, were eliminated, which can reduce service operation costs and operating personnel.

The MSF proposed in this study is a framework applied to the field of business management, as mentioned in Section 3.1. Therefore, future advancements must include the extension of the framework to the field of battlefield management.

**Author Contributions:** Writing—Original draft preparation, S.C.; Project administration, S.H.; Writing—Review and editing, W.S. and N.K.; Supervision, H.P.I. All authors have read and agreed to the published version of the manuscript.

**Funding:** This work was supported by Institute of Information & communications Technology Planning & Evaluation (IITP) grant funded by the Korea government (MSIT) (No.2020-0-00418, Development of Smart Contract Visualization Platform for User Convenience).

**Conflicts of Interest:** The authors declare no conflict of interest.

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
