# Peer review of "Design of Military Service Framework for Enabling Migration to Military SaaS Cloud Environment"

_electronics, doi:10.3390/electronics10050572_

Round 1

Reviewer 1 Report

In this study, a military service framework (MSF) was designed to systematically migrate military applications
to the SaaS cloud environment and manage the SaaS portfolio.
The proposed framework was designed under several preconditions considering the ROK
military's cloud policy.

We list some comments for this manuscript in the following:

  1. What are key ideas in the proposed framework for solving migration to military SaaS cloud? We didn't find authors' designs in the manuscript.
  2. The experimental studies are too weak to support this framework.
  3. The resolution of figures are too low. Most cells in Fig. 3 are blank. Font size is too small to read. The format of table (e.g., the last column of Table 1) needs to be greatly improved.

Reviewer 2 Report

I think that the article does not correspond to the subject of the journal.

In addition, the presentation quality is poor. Why is the text shifted to the right, although the figures seem to be centered.

The purpose of the work and the benefits obtained are not explicitly tracked, except for the illustration in Fig. 10.

Figures 8 and 9 raise questions (especially blank lines). Most likely, the results should be presented differently? Obviously, the results should be made more visible.

Round 2

Reviewer 1 Report

Authors provide new manuscript and solve most issues we proposed.

Reviewer 2 Report

The authors improved the quality of the paper, corrected shortcomings, added the necessary explanations to the text. The paper can be published.